# Economic evaluation of a randomized controlled trial comparing mifepristone and misoprostol with misoprostol alone in the treatment of early pregnancy loss

Charlotte C. Hamel [1,2]*, Marcus P. L. M. Snijders[2], Sjors F. P. J. Coppus[3], Frank P. H. A. Vandenbussche[1,4], Didi D. M. Braat[1], Eddy M. M. Adang[5]

1 Radboud University Medical Centre, Department of Obstetrics and Gynaecology, Nijmegen, GA, The Netherlands, 2 Department of Obstetrics and Gynaecology, Canisius-Wilhelmina Hospital, Nijmegen, GS, The Netherlands, 3 Maxima Medical Centre, Department of Obstetrics and Gynaecology, Veldhoven, DB, The Netherlands, 4 Department of Obstetrics and Gynaecology, Helios Klinikum Duisburg, Duisburg, Germany, 5 Department for Health Evidence, Radboud University Medical Centre, Nijmegen, GA, The Netherlands

* lotte.hamel@radboudumc.nl

**Data Availability Statement:** Confidential data available to researchers who meet the criteria for

## Abstract

### Background

In case of early pregnancy loss (EPL) women can either choose for expectant, medical or surgical management. One week of expectant management is known to lead to spontaneous abortion in approximately 50% of women. Medical treatment with misoprostol is known to be safe and less costly than surgical management, however less effective in reaching complete evacuation of the uterus. Recently, a number of trials showed that prompt treatment with the sequential combination of mifepristone with misoprostol is superior to misoprostol alone in reaching complete evacuation. In this analysis we evaluate whether the sequential combination of mifepristone with misoprostol is cost-effective compared to misoprostol alone, in the treatment of EPL.

### Methods and findings

A cost-effectiveness analysis (CEA) from a healthcare perspective was performed alongside a randomised controlled trial (RCT) in which standard treatment with misoprostol only was compared with a combination of mifepristone and misoprostol, in women with EPL after a minimum of one week of unsuccessful management. A limited societal perspective scenario was added. This RCT, the Triple M trial, was a multicentre, randomized, double-blinded, placebo-controlled trial executed at 17 hospitals in the Netherlands. The trial started on June 27th 2018, and ended prematurely in January 2020 due to highly significant outcomes from the predefined interim-analysis. We included 351 women with a diagnosis of EPL between 6 and 14 weeks gestation after at least one week of unsuccessful expectant management. They were randomized between double blinded pre-treatment with oral mifepristone 600mg (N = 175) or placebo (N = 176) taken on day one, both followed by

access. Data are available from the Radboud UMC Institutional Data Access: Contact via: Radboud UMC Dept. of Obstetrics and Gynaecology Attn. G. Zijdeveld, research coordinator Geert Grooteplein Zuid 10 6525 GA Nijmegen The Netherlands

**Funding:** This study was funded by Healthcare Insurers Innovation Foundation (grant number: 3080 B15–191) awarded to CH. Additional funding, in the form of part of the salary of CH, was provided by Radboud University Medical Centre and Canisius Wilhelmina Hospital.

**Competing interests:** The authors have declared that no competing interests exist.

misoprostol orally. In both groups, an intention-to-treat analysis was performed for 172 patients, showing a significant difference in success rates between participants treated with mifepristone and misoprostol versus those treated with misoprostol alone (79.1% vs 58.7% respectively). In this cost-effective analysis we measured the direct, medical costs related to treatment (planned and unplanned hospital visits, medication, additional treatment) and indirect costs based on the IMTA Productivity Cost Questionnaire (iPCQ). Quality Adjusted Life Years (QALY's) were calculated from participants' scores on the SF-36 questionnaires sent digitally at treatment start, and one, two and six weeks later. We found medical treatment with placebo followed by misoprostol to be 26% more expensive compared to mifepristone followed by misoprostol (p = 0.001). Mean average medical costs per patient were significantly lower in the mifepristone group compared to the placebo group (€528.95 ± 328.93 vs €663.77 ± 456.03, respectively; absolute difference €134.82, 95% CI 50,46–219,18, p = 0.002). Both indirect costs and QALY's were similar between both groups.

## Conclusion

The sequential combination of mifepristone with misoprostol is cost-effective compared with misoprostol alone, for treatment of EPL after a minimum of one week of unsuccessful expectant management.

## Introduction

EPL is defined as a non-viable first trimester intra-uterine pregnancy, in which there may be an anembryonic gestation or embryonic death [1, 2]. It is the most common complication in early pregnancy, with a reported incidence varying from 10% to 28% of pregnancies [3, 4]. The estimated annual number of pregnancies worldwide is 227 million, meaning every year millions of women will seek treatment for EPL.

Three treatment options exist for EPL: expectant, surgical or medical management. In many European countries, including the Netherlands, expectant management of miscarriage (waiting for the miscarriage to occur spontaneously) for at least one week is common practice, as spontaneous complete evacuation occurs in up to 50% of women [5, 6]. However, after this period of expectant management, around half of women experiencing EPL may require treatment. Although very successful in reaching complete evacuation, surgical management, i.e., uterine aspiration, is associated with risks of early and late complications, such as adhesion formation and increased risk of premature delivery in subsequent pregnancies, and higher costs [7–9].

Multiple international guidelines recommend prostaglandins as primary medical treatment of EPL [2, 10]. Misoprostol tablets, a prostaglandin E1 analogue are, although off-label, widely used, relatively cheap, easy to apply and proven safe, not requiring special storage or temperature conditions [2, 10, 11] Medical management using misoprostol without previous expectant management may result in success rates of 66.0% to 88.5% [12, 13]. After one week of unsuccessful expectant management, the success rate of misoprostol treatment drops to around 54% [14, 15]. In short, surgical treatment is associated with risks and higher costs, but medical treatment with misoprostol is limited in terms of efficacy.

Recently, whether or not preceded by expectant management, the sequential combination of mifepristone and misoprostol has been shown to be more successful in reaching complete

expulsion than misoprostol alone in case of EPL as proven by both our and other research groups [16–18].

Here, we present a cost-effectiveness analysis (CEA), performed as a secondary analysis alongside the Triple M Trial, investigating the sequential combination of mifepristone and misoprostol versus placebo followed by misoprostol after at least one week of unsuccessful expectant management in women with EPL.

## Materials and methods

This economic evaluation was performed alongside the Triple M Trial, a nationwide multicentre double-blinded placebo-controlled RCT conducted in 17 Dutch hospitals [16]. A CONSORT checklist and further details about study design, sample size calculation, study procedures and outcome have been described previously [19].

In short, women of at least 16 years of age with confirmed EPL by ultrasonography (at gestational age 6 to 14 weeks), managed expectantly for at least one week without aborting spontaneously, were eligible for inclusion. After written informed consent, women were randomly assigned to either mifepristone 600 mg orally or an identical appearing placebo, containing no active ingredients, followed 36–48 hours later by misoprostol 400 μg 2dd on day three and, if necessary, again on day four.

Successful treatment was defined as ultrasonographic confirmed expulsion of the gestational sac and an endometrial thickness <15 mm after a maximum of 6 weeks after treatment, using only the allocated therapy.

Both the mifepristone and identical appearing placebo tablets were purchased from the same manufacturer, Exelgyn (Groupe Nordic Pharma, France). Exelgyn had no further role in the design or execution of the study, nor in the analysis of study results.

### Ethical approval

Ethical approval for the Triple M Trial was obtained from the Medical Research Ethical Committee region Arnhem-Nijmegen and the National Central Committee on Research involving Human Subjects (file number NL 62449.091.17) In addition, the board of directors of each of the participating centres gave approval to conduct this trial on their respective locations.

### Economic evaluation

Cost-effectiveness was evaluated from a health care perspective. In an additional scenario cost-effectiveness was determined from a societal perspective. Costs, usually skewed, were analysed by a generalized linear model with a log link relating the conditional mean to the treatment dummy using a gamma distribution specifying the relationship between the variance and the mean. The choice for a log link-based model, besides flexibility, was mainly made to present cost difference as a percentage. By doing so, the result is more generalizable to other countries with different healthcare cost systems, as absolute cost figures might differ, relative cost figures may be more insightful.

Absolute cost figures are also presented. Unit costs for outpatient visits, ultrasonography, hospital admission, hysteroscopy, uterine aspiration and packed cells were provided by the financial department of the Radboud University Medical Centre, Nijmegen, the Netherlands. Costs for medication were derived from the Dutch Formulary for medication (https://www.therapeutischkompas.nl (accessed 8 April 2020)), to deliver costs in daily practice. All costs are expressed in Euros, direct costing data were available from all 344 participants.

Indirect costing data about productivity loss were obtained by means of the IMTA Productivity Cost Questionnaire (iPCQ).

Health related quality of life was based on the participants scores on the Dutch version of the RAND-36 questionnaires, measured at four moments: at treatment start, after one, two and six weeks. If the questionnaire was not started at baseline, or the next three questionnaires were not or only barely completed (<20%), these participants were excluded from the quality-of-life analysis. For the remaining questionnaires, missing data were imputed using the 'last observation carried forward' (LOCF) method, a conservative method of data imputation when outcomes are expected to improve over time [20], as expected in these medical circumstances.

Then, according to the Short Form Six Dimension (SF-6D) health state classification form, a preference-based index was deduced [21]. Quality adjusted life years (QALYs) were thus calculated over the period of evaluation using the SF-6D index scores (utilities) multiplied by the consecutive time periods applying the trapezium method. This method evaluates the area under the curves by dividing the total area into smaller trapezoids rather than using rectangles, with the additional advantage of applying correction at each moment of measurement [22].

Incremental cost-effectiveness ratios (ICERs) were calculated generating 95% CIs from 1000 bootstrapped replications with replacement to test the robustness of our cost-effectiveness result. These results are presented in a Cost-Effectiveness-plane (CE plane), graphically illustrating costs and effects of an intervention, and in a Cost-Effectiveness Acceptability Curve (CEAC) summarizing the impact of uncertainty on the economic evaluation.

## Results

### Participants

Between June 27th 2018 and January 8th 2020, a total of 351 patients were enrolled and randomized to either pre-treatment with mifepristone (175 patients) or placebo (176 patients) prior to misoprostol after at least one week of unsuccessful expectant management. Based on the superiority of one of the treatments, the study was halted prematurely, as advised by the trials Data Safety Monitoring Board. After excluding patients who either withdrew their consent, were lost to follow-up or in retrospect did not meet inclusion criteria (one patient turned out to have a cornual pregnancy) 172 participants remained in each group, see Fig 1.

### Clinical effectiveness and QALYs outcomes

Baseline characteristics were comparable between the two groups, as shown in Table 1. The percentage of patients reaching complete evacuation of the uterus without additional treatment (the primary outcome) was significantly higher in the mifepristone group than in the placebo group (79.1% and 58.7% respectively), p = 0.000. Additionally, the number of uterine aspirations performed to achieve complete evacuation was significantly lower in the mifepristone group (19/172 participants) than in the placebo group (51/172 participants), p = 0.000, see also Table 1.

The RAND-36 questionnaire scores were available from 182 participants (94 in the mifepristone and 88 in the placebo group), with again no significant differences in the basic characteristics. These scores were analysed, showing that SF-6D utility index scores did not differ significantly between groups on either measuring moment, see Table 2. The mifepristone group had, over the six weeks period of evaluation, a mean QALY of 0.0853 (95% CI 0.0820–0.0890) compared to a mean QALY of 0.0860 (95% CI 0.0833–0.0892) in the placebo group (p = 0.775).

### Costs and cost-effectiveness

Cost analysis (health care perspective) showed that the mean (± SD) direct, medical costs were significantly lower in the mifepristone group compared to the placebo group (Table 3:

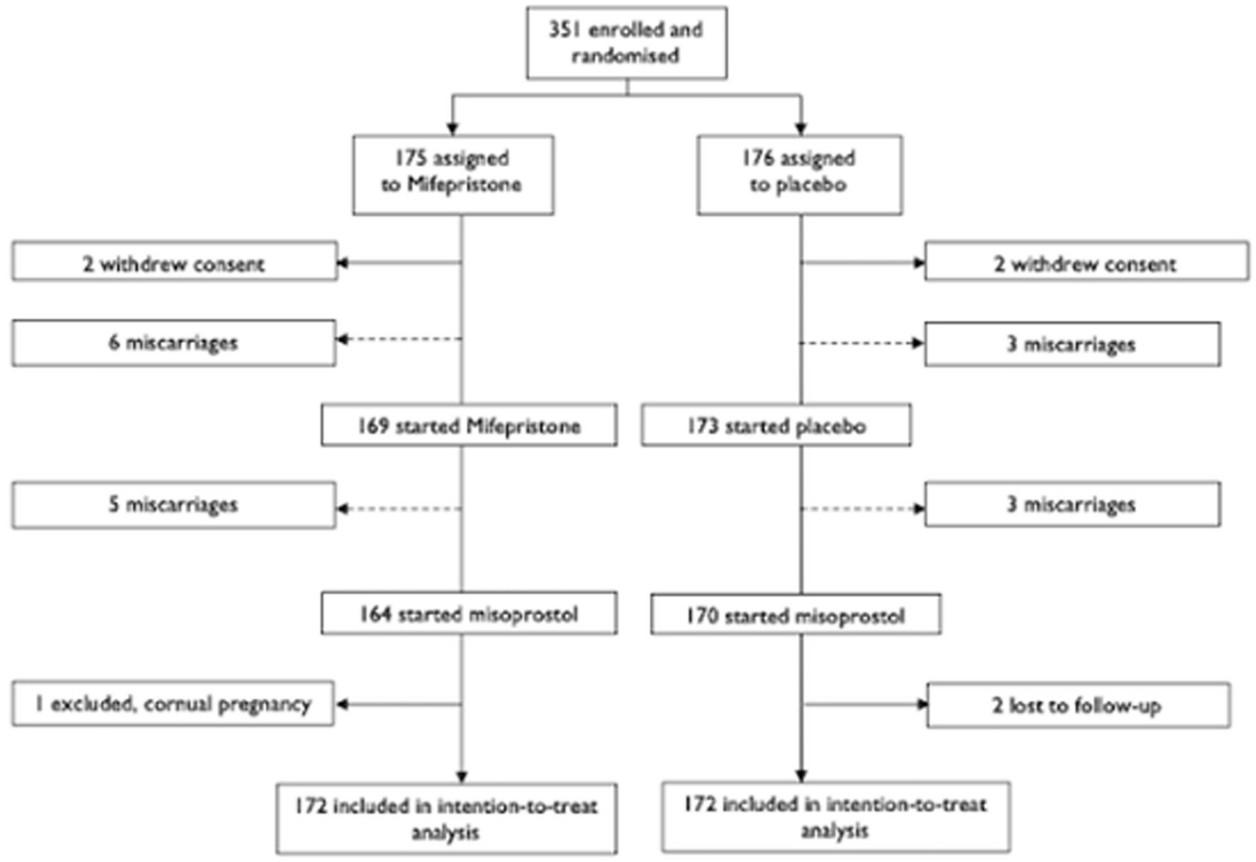

**Fig 1. Trial profile.** ⇢ = included in intention-to-treat analysis. → = excluded from intention-to-treat analysis.

€528.95 ± 328.93 vs €663.77 ± 456.03, respectively; absolute difference €134.82, 95% CI 50,46–219,18, p = 0.002). Medical treatment with placebo followed by misoprostol, analysed by the log link-based model, is 26% more expensive compared to mifepristone followed by misoprostol (p = 0.001).This shows that the overall cost reduction achieved by pre-treatment with mifepristone outweighs the costs of mifepristone tablets.

Total indirect costs from productivity loss, based on scores of the fully completed IPCQ questionnaire were available for 116 participants, 59 in the mifepristone and 57 in the placebo group. Basic characteristics were again similar between these groups. Mean (± SD) indirect costs were comparable between both groups, being €2155.59 ± 3371.29 in the mifepristone group, compared to €2161.70 ± 3100.94 in the placebo group (p = 0.992) and consequently did not alter the cost-effectiveness outcome.

The bootstrapped Incremental Cost-effectiveness ratios (ICERs), or costs per QALY gained, are plotted on the cost-effectiveness (CE) plane in Fig 2, with the control strategy in the origin. This shows that the experimental strategy, i.e. pre-treatment with mifepristone prior to misoprostol, is less costly compared to the standard strategy, i.e. misoprostol only, illustrated by a negative difference in costs. Regarding QALY's the effect is equal for both strategies (differences in effect are not mainly positive nor negative). As the experimental strategy is cheaper but equally effective, it is therefore cost-effective.

From Fig 2 a cost-effectiveness acceptability curve (CEAC) can be deduced, illustrated in Fig 3. The probability that the experimental strategy is cost-effective, compared to the control,

**Table 1. Basis characteristics and clinical outcomes.**

| Characteristic | | Mifepristone and Misoprostol N = 172 | Placebo and Misoprostol N = 172 |
|---|---|---|---|
| Age (years) | | | |
| Mean (SD) | | 32.95 (4.39) | 32.69 (4.30) |
| BMI | | | |
| Mean (SD) | | 24.70 (4.44) | 24.08 (3.84) |
| Unknown | | 28 | 34 |
| Race or ethnic group | | | |
| Caucasian | | 156 (90.7%) | 155 (90.1%) |
| Other | | 12 (7.0%) | 13 (7.6%) |
| Unknown | | 4 (2.3%) | 4 (2.3%) |
| Gravidity | | | |
| | 1 | 60 (34.9%) | 75 (43.6%) |
| | 2 | 63 (36.6%) | 53 (30.8%) |
| | ≥3 | 49 (28.5%) | 44 (25.6%) |
| Parity | | | |
| | 0 | 83 (48.3%) | 94 (54.7%) |
| | 1 | 70 (40.7%) | 64 (37.2%) |
| | ≥2 | 19 (11.0%) | 14 (8.1%) |
| Gestational age based on amenorrhoea (days) | | | |
| Mean (SD) | | 71.22 (11.03) | 70.09 (11.57) |
| Unknown | | 3 | 3 |
| Diagnosis | | | |
| Embryo without cardiac activity | | 123 (71.5%) | 115 (66.9%) |
| Anembryonic gestation | | 49 (28.5%) | 57 (33.1%) |
| Prior miscarriage | | 51 (29.7%) | 52 (30.2%) |
| Of these: misoprostol treatment for prior miscarriage | | 13 (25.5%) | 19 (36.5%) |
| Of these: successful misoprostol treatment | | 7 (53.8%) | 12 (63.2%) |
| **Clinical outcomes** | | | |
| Complete evacuation | | 136 (79.1%) | 101 (58.7%) |
| Uterine aspiration | | 19 (11.0%) | 51 (29.7%) |
| Other additional therapy | | 17 (9.9%) | 20 (11.6%) |

decreases if the willingness to pay (WTP) increases, due to a small but insignificant difference between the point estimates of the SF-6D score in the control and experimental group.

Over the relevant range, with a maximum of €80.000 as additional costs per QALY, the probability that the experimental strategy is more cost-effective than the control strategy is higher.

Uncertainty of these probabilities increases when the WTP for a QALY increases. This is shown in Fig 4; with the EVPI (Expected Value of Perfect Information, the price that one would be willing to pay in order to gain access to perfect information) increasing with increasing WTP. This shows that when the WTP increases, perfect information becomes more valuable (i.e. the EVPI rises), to counter the increased uncertainty. At a WTP of €80.000, the EVPI is about €55 per patient. On the other hand, if one only focuses on cost the EVPI is near €0, as cost are significantly lower in the experimental group, making the uncertainty surrounding the cost driven cost-effectiveness decision rule vanish.

**Table 2. SF-6D utility index scores per measuring moment in 1000 bootstrapped simulations.**

| | Mean utility scores (± SD) | | |
|---|---|---|---|
| | **Mifepristone and misoprostol (N = 94)** | **Placebo and misoprostol (N = 88)** | **P-value** |
| **Measuring moment** | | | |
| Treatment start (T = 1) | 0.7387 (± 0.097) | 0.7346 (± 0.109) | 0.783 |
| One week after treatment start (T = 2) | 0.6713 (± 0.109) | 0.6938 (± 0.119) | 0.187 |
| Two weeks after treatment start (T = 3) | 0.6852 (± 0.102) | 0.6979 (± 0.115) | 0.420 |
| Six weeks after treatment start (T = 4) | 0.7416 (± 0.119) | 0.7609 (± 0.106) | 0.232 |

## Discussion

This cost-effectiveness analysis shows that, after one week of unsuccessful expectant management, pre-treatment with mifepristone 600 mg taken orally, prior to misoprostol 400 μg 2dd taken orally, for one or two days, is a cost-effective alternative to misoprostol alone as medical management for EPL. Overall, treatment with misoprostol only was found to be 26% more expensive compared to treatment with mifepristone and misoprostol (experimental strategy, p = 0.001). From a societal perspective, indirect costs were comparable between both groups. Although the experimental treatment has slightly higher medication costs, the increase in clinical effectiveness results in a less expensive treatment regimen overall, with a difference of € 134,80.

The key strength of this cost-effectiveness analysis is that it was performed alongside a well-designed RCT, enabling firm conclusions. Additionally, as we opted for a generalized linear model with a log link, comparison with other cost-effectiveness analyses on this treatment regimen is facilitated. A limitation might be that QALYS and indirect costs could not be calculated for all participants, as not all participants completed the questionnaires. However, with a response rate of 66% we believe solid conclusions can be drawn from these data. Furthermore, it is important to realize that LOCF as an imputation method, which was used here, can be prone to bias, depending on the distribution of the observed values. However, as the pattern over time in the experimental and control group is comparable before and after imputation, and the trend observed after LOCF is similar compared to that in the pilot study of our RCT, this imputation method appears suitable for this analysis.

**Table 3. Direct costs for completed treatment per patient.**

| | Mean direct, medical costs per patient (±SD); € | | | |
|---|---|---|---|---|
| | **Mifepristone and misoprostol (N = 172)** | **Placebo and misoprostol (N = 172)** | **Absolute between-group difference (95% CI)** | **P-value*** |
| ***Direct, medical costs*** | ***528.95 (± 328.93)*** | ***663.77 (± 456.03)*** | ***134.82 (50.46–219.18)*** | ***0.002*** |
| Medication | 71.27 (± 8.95) | 37.76 (± 11.91) | 33.50 (35.73–31.27) | **0.000** |
| Ultrasound | 150.12 (± 50.32) | 151.14 (± 45.63) | 1.02 (-9.17–11.21) | 0.844 |
| Hospital visits | 179.35 (± 12.93) | 179.44 (± 12.23) | 0.09 (-2.58–2.76) | 0.946 |
| Hospital admissions | 46.28 (± 136.20) | 104.13 (± 179.70) | 57.85 (24.03–91.68) | **0.001** |
| Uterine aspiration | 64.92 (± 184.76) | 174.26 (± 269.19) | 109.34 (60.35–158.33) | **0.000** |
| Hysteroscopy | 16.94 (± 64.99) | 13.86 (± 59.15) | 3.08 (-16.26–10.10) | 0.646 |
| Packed cells | 0 (± 0) | 3.10 (± 40.66) | 3.10 (-3.02–9.22) | 0.319 |
| Antibiotics | 0.079 (± 1.04) | 0.079 (± 1.04) | 0.00 (-0.22–0.22) | 1.000 |

Plus-minus values are mean ± SD.

*Student's t-test; a P-value <0.05 was considered statistically significant.

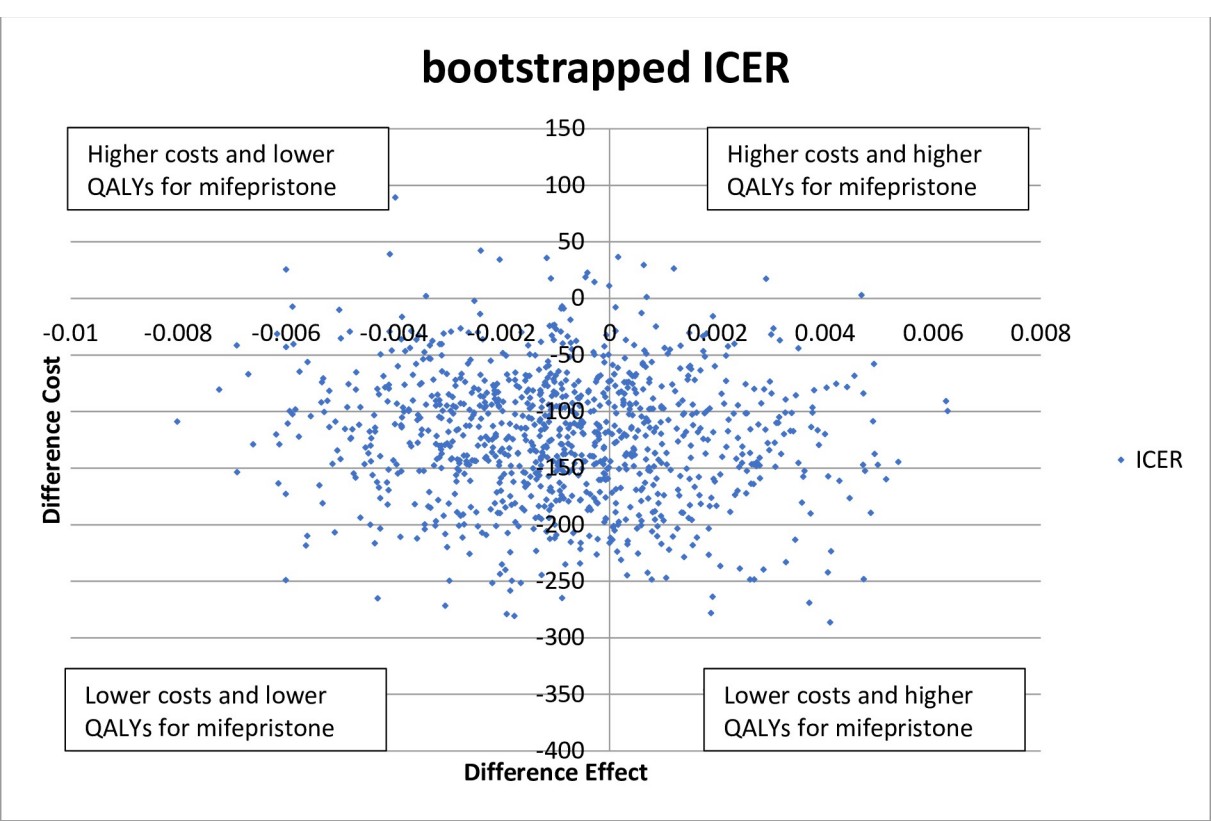

**Fig 2. Cost-effectiveness plane: Scatter plot showing the mean difference in cost per QALY gained i.e. incremental cost-effectiveness, in 1000 bootstrapped simulations, (standard strategy in origin).**

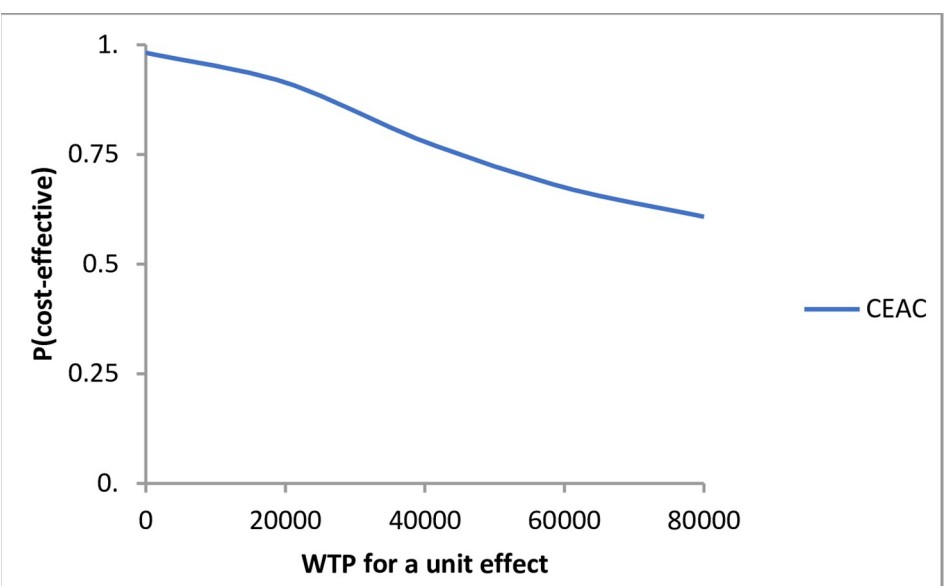

**Fig 3. Cost effectiveness acceptability curve.**

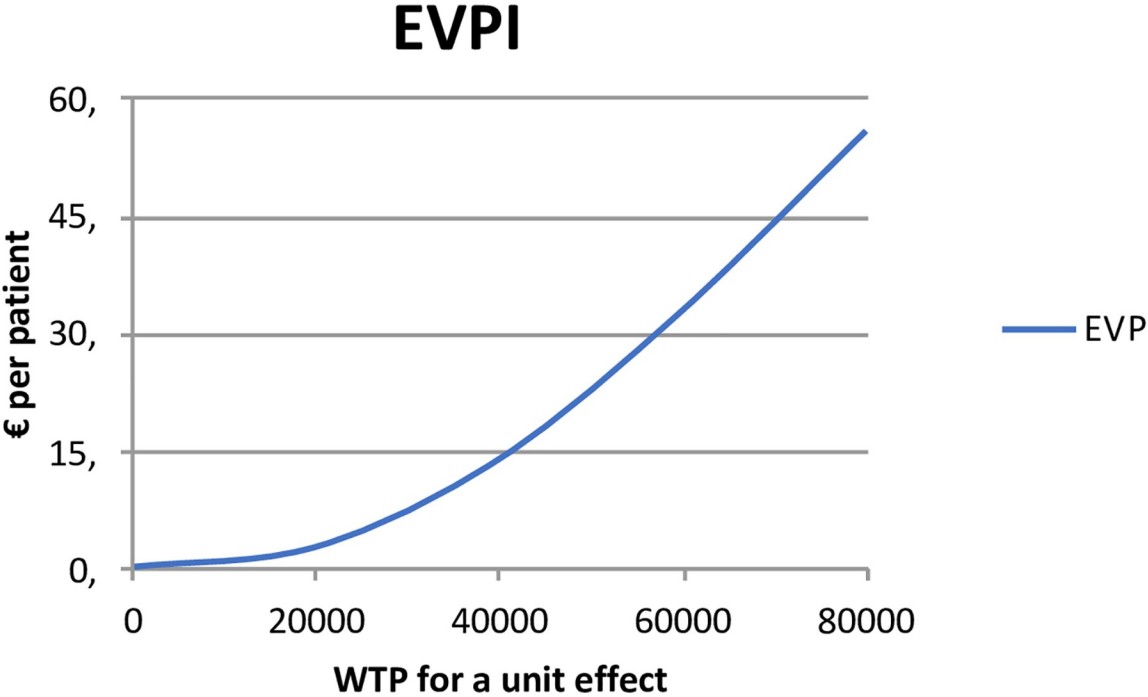

**Fig 4. Expected value of perfect information curve.**

Recently Nagendra et al. [23], reported on a cost-effectiveness analysis, based on a pragmatic comparative effectiveness trial in the United States. In this trial, prompt treatment, without expectant management in case of EPL, was applied. Treatment consisted of 200 mg mifepristone followed by 800 mcg misoprostol vaginally, or 800 mcg misoprostol vaginally only, without prior use of a placebo. Successful treatment was defined as loss of the gestational sac at 24 to 96 hours after misoprostol use, and no need for surgical intervention in the following 30 days. Nagendra et al. also concluded that pre-treatment with mifepristone is the cost-effective alternative in case of EPL. This was also the conclusion of a secondary analysis of their data, in which Monte Carlo simulations were used to assess contribution of different expense categories [24]. An in-depth comparison between these two trials may be difficult as study design varies substantially, regarding medication regimen, applying expectant management prior to inclusion or prompt treatment and mean gestational age, which is approximately three weeks higher in our study. Of course, the healthcare system is arranged quite different in the United States compared to the Netherlands, especially regarding costs.

Comparing both studies, direct costs were significantly lower in the mifepristone group compared with the placebo group in this Triple M trial. In contrast to Nagendra et al., uterine aspiration, hospital admission as well as pharmaceutical costs were all significantly different between both treatment groups in our analysis. A significantly higher number of uterine aspirations, automatically involving hospital admission in the Netherlands, led to higher costs for the misoprostol only group in this study. This may be partly due to the fact that in the Netherlands uterine evacuation is usually performed in a clinical setting, accompanied by hospital admission, whereas in the United states manual vacuum aspiration (MVA) in an outpatient setting may be more common, thus being potentially cheaper [25, 26]. Regarding pharmaceutical costs, in two other studies showing a therapeutical advantage of prompt pre-treatment with mifepristone in case of EPL a dosage of 200 mg mifepristone was used, potentially leading

to less direct pharmaceutical costs [17, 18]. In our RCT after at least one week of unsuccessful expectant management a dosage of 600 mg mifepristone was chosen for three reasons. Firstly, non-viable pregnancies may require a higher dosage of mifepristone then is used in medical abortion. Secondly, as these pregnancies still remain intra-uterine, even after a minimum of one week of expectant management, a higher dose may be needed. In our Triple M Trial, the average gestational age Thirdly, a higher dosage of mifepristone does not necessarily lead to more side-effects, and has even been found to cause less pain in case of medical abortion of viable first-trimester pregnancy [27].

As the patent of mifepristone has expired, pharmaceutical costs are more and more reduced. In fact, during this Triple M trial, costs for Mifepristone were €11,99 for one tablet of 200 mg, compared to reported $90,—in the PreFair trial by Schreiber et al. Future dosage finding studies for mifepristone as well as misoprostol may provide definitive clarity about optimal dosage for efficacy, regarding prior expectant management yes or no, adverse reactions, quality of life, and costs.

Further comparison of both CEA's reveals that, although the maximum WTP is $150,000 for the United States [28], against €80.000 (approximately $94,000) in the Netherlands [29], both CEA's conclude that over the relevant range (for the country involved where each trial was performed), the probability that mifepristone pre-treatment is cost-effective is highest.

In the Triple M Trial expectant management around EPL treatment was deliberately chosen as a preferred policy. Apart from a minimum of one week of expectant management, we asked participants, taking their preference into account, to return six weeks after treatment start if ultrasound at the first follow-up at two weeks was suspect for retained products of conception (RPOCs) i.e. gestational sac expulsed, but endometrial thickness > 15 mm [30]. This posttreatment expectant policy is based on three pillars. Firstly, it has been shown that an expectant policy in case of suspected RPOC after misoprostol treatment for EPL is equally safe compared to prompt uterine aspiration, thus preventing additional treatment in the majority (up to 85%) of women [31]. Secondly, uterine evacuation has not been proven cost-effective compared with expectant management in this scenario [32]. Finally, it has been shown that, in women in whom expectant management was applied in case of suspected RPOC after misoprostol treatment for EPL, health-related quality of life improved more and earlier than of women who underwent surgical evacuation [33]. This preferred expectant policy and thus restricted use of additional therapy may lead to less interventions, resulting in lower costs, without negatively influencing health-related quality of life and also crucial, accepting patient's preference. However, we realize that in other settings prompt treatment may be preferred, as expectant management can only be applied under the condition that a solid and easily accessible (24/7) backup setting for emergencies is available. This may not be the case in all healthcare systems worldwide, thus influencing feasibility and cost effectiveness.

In conclusion, pre-treatment with mifepristone prior to misoprostol in case of EPL after a minimum of one week of unsuccessful expectant management not only increases success-rates, but is also cost-effective compared to misoprostol alone. This will have important implications worldwide as both mifepristone and misoprostol, not requiring special storage conditions, become more and more freely available and affordable in terms of pharmaceutical costs. These features, combined with an overall lower risk of complications argue for a more wide-spread use of, pre-treatment with mifepristone in case of EPL whether or not after a period of expectant management in both high- and low-income countries. Future dose-finding studies might provide definitive clarity about the optimal dosage of mifepristone, in which costs should also be taken into account.

## Author Contributions

**Conceptualization:** Marcus P. L. M. Snijders, Sjors F. P. J. Coppus, Frank P. H. A. Vandenbussche.

**Data curation:** Charlotte C. Hamel.

**Formal analysis:** Charlotte C. Hamel, Eddy M. M. Adang.

**Funding acquisition:** Marcus P. L. M. Snijders.

**Investigation:** Charlotte C. Hamel.

**Supervision:** Marcus P. L. M. Snijders, Eddy M. M. Adang.

**Writing – original draft:** Charlotte C. Hamel, Marcus P. L. M. Snijders, Sjors F. P. J. Coppus, Frank P. H. A. Vandenbussche, Eddy M. M. Adang.

**Writing – review & editing:** Charlotte C. Hamel, Marcus P. L. M. Snijders, Sjors F. P. J. Coppus, Frank P. H. A. Vandenbussche, Didi D. M. Braat, Eddy M. M. Adang.

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
