## [Decision Letter · Decision Letter 0]

11 Aug 2021

PONE-D-21-02362

Economic evaluation of a randomized controlled trial comparing mifepristone and misoprostol with misoprostol alone in the treatment of early pregnancy loss.

PLOS ONE

Dear Dr. Hamel,

Thank you for submitting your manuscript to PLOS ONE. After careful consideration, we feel that it has merit but does not fully meet PLOS ONE’s publication criteria as it currently stands. Therefore, we invite you to submit a revised version of the manuscript that addresses the points raised during the review process.

The reviewers had concerns about some of the statistical analyses and the presentation of some results. Their comments can be viewed in full, below.

We look forward to receiving your revised manuscript.

Kind regards,

Natasha McDonald, PhD

Associate Editor

PLOS ONE

Journal Requirements:

3. For more information on PLOS ONE's expectations for statistical reporting, please see https://journals.plos.org/plosone/s/submission-guidelines.#loc-statistical-reporting. Please update your Methods and Results sections accordingly.

4. Please update your submission to use the PLOS LaTeX template. The template and more information on our requirements for LaTeX submissions can be found at http://journals.plos.org/plosone/s/latex.

5. We noted in your submission details that a portion of your manuscript may have been presented or published elsewhere. EClinicalMedicine (published by the Lancet group). Please clarify whether this publication was peer-reviewed and formally published. If this work was previously peer-reviewed and published, in the cover letter please provide the reason that this work does not constitute dual publication and should be included in the current manuscript.

6. Please upload a new copy of Figure 1 as the detail is not clear. Please follow the link for more information: " ext-link-type="uri" xlink:type="simple">https://blogs.plos.org/plos/2019/06/looking-good-tips-for-creating-your-plos-figures-graphics/"
" ext-link-type="uri" xlink:type="simple">https://blogs.plos.org/plos/2019/06/looking-good-tips-for-creating-your-plos-figures-graphics/"

Reviewers' comments:

Reviewer's Responses to Questions

**Comments to the Author**

1. Is the manuscript technically sound, and do the data support the conclusions?

Reviewer #1: Yes

Reviewer #2: Yes

Reviewer #3: Yes

2. Has the statistical analysis been performed appropriately and rigorously? 

Reviewer #1: Yes

Reviewer #2: Yes

Reviewer #3: Yes

3. Have the authors made all data underlying the findings in their manuscript fully available?

Reviewer #1: Yes

Reviewer #2: Yes

Reviewer #3: Yes

4. Is the manuscript presented in an intelligible fashion and written in standard English?

Reviewer #1: Yes

Reviewer #2: Yes

Reviewer #3: No

5. Review Comments to the Author

Reviewer #1: This is a well conducted secondary analysis of the cost-effectiveness data for the Triple-M trial. This study was pre-registered and all outcomes were reported as specified. I only have one minor comment, which is that the last sentence of the discussion is incomplete 'Future dose-finding studies might'.

Reviewer #2: Important note: This review pertains only to ‘statistical aspects’ of the study and so ‘clinical aspects’ [like medical importance, relevance of the study, ‘clinical significance and implication(s)’ of the whole study, etc.] are to be evaluated [should be assessed] separately/independently. Further please note that any ‘statistical review’ is generally done under the assumption that (such) study specific methodological [as well as execution] issues are perfectly taken care of by the investigator(s). This review is not an exception to that and so does not cover clinical aspects {however, seldom comments are made only if those issues are intimately / scientifically related intermingle with ‘statistical aspects’ of the study}. Agreed that ‘statistical methods’ are used as just tools here, however, they are vital part of methodology [and so should be given due importance].

COMMENTS: In my opinion, it may be useful to add [for general / non-subject expert readers] that ‘Expectant management’ means waiting for the miscarriage to happen by itself naturally, without treatment]. Since misoprostol orally is common to both it is perfectly alright to administer ‘mifepristone 600mg (N=175)’ to one group and its ‘placebo (N=176)’ to other, however, more details of ‘placebo’ are missing [mention that ‘an identical appearing placebo’ may not suffice]. Moreover, your saying “medical treatment with placebo followed by misoprostol to be 26% more expensive compared to mifepristone followed by misoprostol (p=0.001)” is not very clear because the difference is ‘mifepristone 600mg’ versus its ‘placebo’ which implies that ‘placebo’ is more expensive [how medical fraternity or patients will benefit by such info?]. If you intend to convey something else, mention that clearly.

Description/details regarding many of the Important items in CONSORT checklist [Examples: Allocation concealment (item 9), Blinding (item 11a)] are not found. This trial is said to be ‘double-blinded placebo-controlled RCT’ but no blinding details included. The word ‘CONSORT’ itself is surprisingly not found in the article. The economic evaluation was performed correctly but there is hardly anything on other analytical {mainly statistical} methods used. Study protocol for this trial is published separately [BMC Pregnancy Childbirth. 2019;19(1):1–8] is good, however, few details about study design, sample size calculation, study procedures and outcome should have been briefly described here also. As you clarified, you are presenting a cost-effectiveness analysis (CEA), [performed as a secondary analysis], but note that this one is separate article (and not a part series), therefore, few details are expected/mandatory I guess.

Statistical comparison of baseline characteristics [last ‘p-value’ column in Table 1] is not desirable at all. In this context, read the following:

To provide a description of baseline characteristics is entirely reasonable (since it is clearly important in assessing to whom the results of the trial can be applied), however, it does not require the division of baseline characteristics by treatment groups (however, if done – alright). Statistical comparison of baseline characteristics is not desirable at all [because even if P-value turns out to be significant (while comparing baseline characteristics despite random allocation), it is, by definition, a false positive] as you then are supposed to be testing ‘randomization’ then, which in any single trial may not balance all baseline characteristics because ‘randomization’ is a sort of ‘insurance’ and not a guarantee scheme.

References:

1. Stuart J. Pocock, et al., ‘Subgroup analysis, covariate adjustment and baseline comparisons in clinical trial reporting: current practice and problems’, Statistics in medicine, 2002; 21:2917–2930 [Particularly page 2927]

2. Harrington D, et al., ‘New guidelines for statistical reporting in the journal’, N Engl J Med 2019;381:285-6

[Important message (indirectly/ultimately indicated) from these articles: Never do any comparison with respect to ‘baseline’ characteristics {by applying statistical significance test(s)}, when allocation is done randomly].

In last ‘p-value’ column in Table 1, there are three (or more) ‘n.s.’ appearing with respect to few variables/characteristics [ex.: Gravidity], which implies that each row is tested separately. Is that correct? Which test is applied for table-2 data? Though the measures/tools used are appropriate, most of them [including SF-36 {as you said: Quality Adjusted Life Years (QALY’s) were calculated from participants’ scores on the SF-36 questionnaires sent digitally at treatment start, and one, two and six weeks later} yield data that are in [at the most] ‘ordinal’ level of measurement [and not in ratio level of measurement for sure {as the score two times higher does not indicate presence of that parameter/phenomenon as double (for example, a Visual Analogue Scales VAS score or say ‘depression’ score)}]. Then application of suitable non-parametric test(s) is/are indicated/advisable [even if distribution may be ‘Gaussian’ (i.e. normal)]. Agreed that there is/are no non-parametric test(s)/technique(s) available to be used as alternative in all situation(s) [suitable / most desired/applicable], but should be used whenever/wherever they are available.

Since missing data were imputed using the ‘last observation carried forward’, I hope the authors are aware of disadvantages [like this method assumes that the response remains constant at the last observed value. This assumption can be biased if the timing and the rate of withdrawal is related to the treatment (e.g. in the case of degenerative diseases, using the last observed value to impute for missing data at a later point in the study means that a higher observation will be carried forward, resulting in an overestimation of the true end-of study measurement)] of the method (must be known to these learned authors, still may please be noted) that:

“according to available literature [example, “Inference and Missing Data,” Biometrika, 1976, vol:63, 581–592 and “Multiple Imputation After 18+ Years,” Journal of the American Statistical Association, 1996, vol:91, 473–489] ‘Multiple Imputation’ technique is preferred [considering MCAR (Missing Completely At Random) expected nature of data] than {despite being time-consuming and involving much more computations} compare to all out of other important imputation techniques frequently used [like Group Means, Hot-deck Imputation, Baseline Observation Carried Forward (BOCF), Worst Observation Carried Forward (WOCF), Predicted Mean, and even Last Observation Carried Forward (LOCF)].”

Except these minor points, the article is excellent [by all the means]. It is definitely in ‘acceptable’ category. Very good job. Appreciated.

Reviewer #3: This is a well presented manuscript presenting the cost effectiveness analysis of a robustly performed multi-centre randomised controlled trial - Triple M - from the Netherlands. The trial was curtailed prematurely due to the findings of the interim data analysis due to clear benefit seen in women receiving mifepristone in combination with misoprostol in the medical management of early pregnancy loss (EPL).

This study shows that mifepristone and misoprostol incurs a significantly lower cost when compared to misoprostol alone. A cost saving of 26%. The authors have demonstrated that the main reasons for reducing cost if a lower cost of medications, admissions and a reduced need for uterine aspiration.

There are some improvements that could be made to the manuscript though:

I would suggest that the editorial board review the manuscript to aid the authors improve English language. There are grammatical and stylistic issues throughout the manuscript. There are also some scientific writing issues in the manuscript such as the use of abbreviations within the abstract of the manuscript and also the use of a question within the background of the abstract. Another example, would be the abbreviation of EPL at the beginning of the abstract which had not yet been explained.

The introduction is excellent and sets the scene very well. The authors have acknowledged the other trials and cost-effectiveness analyses performed on this research question that have been published before.

The method section is well written and the tools used for economic evaluation are well described and well established in health economic evaluation research.

The results section demonstrate the cost benefits clearly and the fact that there are no real differences in the two trial arms when measuring societal costs.

The discussion makes appropriate deductions from the data provided in the results section. The end of the discussion does seem to be incomplete and the authors will need to rectify this.

6. PLOS authors have the option to publish the peer review history of their article (what does this mean?). If published, this will include your full peer review and any attached files.

Reviewer #1: No

Reviewer #2: No

Reviewer #3: **Yes: **Justin Chu

---

## [Author Response · Author response to Decision Letter 0]

20 Sep 2021

For our elaborate response to reviewers please see the attached file 'Response to Reviewers'. 

We thank all reviewers for their time and effort, and overall kind words.

---

## [Decision Letter · Decision Letter 1]

10 Jan 2022

Economic evaluation of a randomized controlled trial comparing mifepristone and misoprostol with misoprostol alone in the treatment of early pregnancy loss.

PONE-D-21-02362R1

Dear Dr. Hamel,

We’re pleased to inform you that your manuscript has been judged scientifically suitable for publication and will be formally accepted for publication once it meets all outstanding technical requirements.

Kind regards,

Stefan Gebhardt, MB CHB, MMedSci, MSc, FCOG, MMed (OG), PhD

Academic Editor

PLOS ONE

Additional Editor Comments (optional):

Reviewers' comments:

Reviewer's Responses to Questions

**Comments to the Author**

1. If the authors have adequately addressed your comments raised in a previous round of review and you feel that this manuscript is now acceptable for publication, you may indicate that here to bypass the “Comments to the Author” section, enter your conflict of interest statement in the “Confidential to Editor” section, and submit your "Accept" recommendation.

Reviewer #2: All comments have been addressed

Reviewer #3: All comments have been addressed

2. Is the manuscript technically sound, and do the data support the conclusions?

Reviewer #2: (No Response)

Reviewer #3: Yes

3. Has the statistical analysis been performed appropriately and rigorously? 

Reviewer #2: (No Response)

Reviewer #3: Yes

4. Have the authors made all data underlying the findings in their manuscript fully available?

Reviewer #2: (No Response)

Reviewer #3: Yes

5. Is the manuscript presented in an intelligible fashion and written in standard English?

Reviewer #2: (No Response)

Reviewer #3: Yes

6. Review Comments to the Author

Reviewer #2: COMMENTS: All of the comments made on earlier draft by me (and hopefully by other respected reviewers also) were/are attended [though I had suggested minor points and mentioned that the article is excellent by all the means]. I recommend the acceptance as the manuscript now (even earlier I accepted/appreciated the potential of this article) has achieved acceptable level, in my opinion.

Reviewer #3: Nil. All of my suggestions have ben addressed and so no further changes are required before the manuscript can be published in PLOS one.

7. PLOS authors have the option to publish the peer review history of their article (what does this mean?). If published, this will include your full peer review and any attached files.

Reviewer #2: **Yes: **Dr. Sanjeev Sarmukaddam

Reviewer #3: **Yes: **Justin Chu

---

## [Editor Report · Acceptance letter]

27 Jan 2022

PONE-D-21-02362R1 

Economic evaluation of a randomized controlled trial comparing mifepristone and misoprostol with misoprostol alone in the treatment of early pregnancy loss. 

Dear Dr. Hamel:

I'm pleased to inform you that your manuscript has been deemed suitable for publication in PLOS ONE. Congratulations! Your manuscript is now with our production department. 

Kind regards, 

on behalf of

Prof Stefan Gebhardt 

Academic Editor

PLOS ONE